# Left Ventricular Non-Compaction Cardiomyopathy-Still More Questions than Answers

**DOI:** 10.3390/jcm11144135

**Published:** 2022-07-16

**Authors:** Jerzy Paluszkiewicz, Hendrik Milting, Marta Kałużna-Oleksy, Małgorzata Pyda, Magdalena Janus, Hermann Körperich, Misagh Piran

**Affiliations:** 11st Department of Cardiology, Poznań University of Medical Sciences, ul. Długa 1/2, 61-848 Poznań, Poland; marta.kaluzna-oleksy@skpp.edu.pl (M.K.-O.); malgorzata.pyda@skpp.edu.pl (M.P.); magdalena.janus@skpp.edu.pl (M.J.); 2Erich und Hanna Klessmann-Institute for Cardiovascular Research and Development, Heart and Diabetes Center NRW, University Hospital of the Ruhr-University Bochum, Georgstr. 11, D-32545 Bad Oeynhausen, Germany; hmilting@hdz-nrw.de; 3Department of Radiology, Nuclear Medicine and Molecular Imaging, Heart and Diabetes Center NRW, University Hospital of the Ruhr-University Bochum, Georgstr. 11, D-32545 Bad Oeynhausen, Germany; hkoerperich@hdz-nrw.de (H.K.); mpiran@hdz-nrw.de (M.P.)

**Keywords:** left ventricular noncompaction, echocardiography, CMR, cardiomyopathy

## Abstract

Left ventricular non-compaction (LVNC) describes the phenotypical phenomena characterized by the presence of excessive trabeculation of the left ventricle which forms a deep recess filled with blood. Considering the lack of a uniform definition of LVNC as well as the “golden standard” it is difficult to estimate the actual incidence of the disease, however, seems to be overdiagnosed, due to unspecific diagnostic criteria. The non-compacted myocardium may appear both as a disease representation or variant of the norm or as an adaptive phenomenon. This article covers different approaches to incidence, pathogenesis, diagnostics, and treatment of LVNC as well as recommendations for patients during follow-up.

## 1. Introduction

Left ventricular non-compaction describes the phenotypical phenomena characterized by the presence of excessive trabeculation of the left ventricle, which forms a deep recess filled with blood. The first autopsy report describing an “embryonic spongy pattern of the left ventricle” in newborns was published in 1926 [1]. The later autopsy reports described single cases of newborns and infants with complex congenital heart disease and specific patterns of the left ventricular muscle. The first case of an isolated sinusoid anomaly of the left ventricle diagnosed with echocardiography by 33 years old woman was published in 1984 [2]. The term “sinusoid anomaly of the left ventricle” was used till the publication of Chin et al. in 1990, who used for the first time the term, “isolated noncompaction of the left ventricular myocardium (INVM)” [3]. At the same time, the authors suggested disturbance of the embryonic compaction process as the cause of the disease and proposed the first echocardiographic diagnostic criteria. With the dynamic development of imaging techniques such as echocardiography, magnetic resonance, and computer tomography, many different phenotypes of “noncompaction” of the left ventricle have been recognized and many definitions have been proposed since the first description [4]. Currently, the term left ventricular noncompaction (LVNC) is used for specific phenotypes of left ventricular anatomy including the presence of prominent trabeculations with deep intertrabecular recesses (non-compacted myocardium) and thin compact layer of the myocardium (compacted myocardium); however, LVNC does not necessarily mean pathology [5]. In some cases, the right ventricle can also be involved. The term LVNC is used by some authors for describing morphological anomalies without impairing LV function and LVNC cardiomyopathy or NCCM (non-compaction cardiomyopathy) for cases with impaired left ventricular function [6]. Not only the general definition of LVNC is disputed, but also its classification as cardiomyopathy differs between the European Society of Cardiology and American Heart Association classification. The first defines LVNC as “non-classified cardiomyopathy”, and the latter as “genetic cardiomyopathy” [7,8]. The MOGE(S) classification proposes a description of left ventricular morphology (M_LVNC_) for the left ventricle without dilatation and preserved systolic function, (M_LVNC D_) for dilated LV, (M_LVNC H_) for hypertrophied LV, and (M_LVNC A_) for arrhythmogenic right ventricular cardiomyopathy [9]. 

Different subtypes classifications were suggested because the LVNC is a very heterogeneous morphological anomaly. Arbustini and colleagues proposed the latter in 2016 [10]. They divided LVNC into seven subtypes (Table 1). Other authors proposed four subtypes of LVNC: Isolated NCCM, NCCM/HCM, NCCM/DCM, and NCCM/HCM/DCM [11]. Most of the cohorts reported in the literature are treated as a whole, without analyzing subgroups, making it difficult to compare individual publications.

Considering the lack of a uniform definition of LVNC, as well as the “golden standard”, it is difficult to estimate the actual incidence of the disease. There are significant differences between the pediatric and adult populations. The incidence of LVNC in children varied according to registers between 4.8 and 9.2% making LVNC the third most common cardiomyopathy [4]. The prevalence in adults is lower, reaching 4.1–5%, and men are almost three times more affected [4]. The familial incidence reaches 40%. A higher degree of trabeculation together with mildly reduced EF was observed in athletes of African ethnicity [12]. Similar results showed the analysis of 1123 participants of the Multi-Ethnic Study of Atherosclerosis (MESA) using CMR fractal analysis [13]. The fractal dimension of 1.3 or more was found more frequently in probands with Afro-American and Hispanic ethnicity. The prevalence also varied between the diagnostic methods used. The LVNC diagnosis appeared up to 0.26% of echocardiographic examinations but reached 14.8% in healthy volunteers examined with MRI [14]. Significant differences exist also using the same diagnostic modality between different patient cohorts. In the metanalysis of Ross et al., the prevalence of LVNC as assessed by echocardiography in healthy controls was 1.05%, in athletic cohorts 3.16%, and in pregnant women up to 18.6% [15]. There are also significant differences in the frequency of incidence of LVNC within one diagnostic method. Using four different diagnostic criteria in the MRI examination, a significant discrepancy in the frequency of diagnosis of LVNC was found. LVNC was diagnosed in 39% of patients using Petersen criteria, 23% using Stacey criteria, 25% with Jacquier, and only 3% using Captur criteria [16]. Furthermore, a median follow-up of 7 years, showed no differences in outcome between patients with and without LVNC regardless of the evaluation method used. 

## 2. Pathogenesis

The pathogenesis of LVNC is not undoubtedly established. The “theory of non-compaction” is the most accepted since Chin used for the first time the term “isolated noncompaction of the left ventricular myocardium” and proposed the possible mechanism [3]. According to this theory, disturbances in the normal development of the heart muscle lead to the persistence of hypertrabeculation at the cost of a thin compact subepicardial layer. In the complex process of human heart development, trabeculations appear in the cardiac jelly at the end of 4 weeks of gestation, forming spongy structures protruding in the ventricular lumen. At this time, coronary circulation does not exist yet, the highly developed trabecular structure of the endocardial part of the ventricular wall ensures sufficient oxygen supply. The thin subepicardial part of the heart muscle forms the “compacted myocardium” while the trabecular part forms the “non-compacted” part. In the next weeks, the trabeculae thicken increasing the volume of the compacted layer while intertrabecular recesses compress and form capillaries [17]. The disturbances of this process, whatever the reason, lead to the persistence of hypertrabeculation and LVNC. However, one can expect that during the compaction process a rapid decrease in the trabeculae volume and an increase in the compacted layer volume should be present. Unfortunately, such a process was documented only in chickens, but not in humans [18,19]. Faber et al. performed a meta-analysis of 31 papers dealing with human ventricle development [20]. Despite inconsistent data and different measurement methods used, it seems to be certain that the growth of the trabeculae is constant throughout the development phase. A decrease in trabeculae volume could not be documented in humans and the authors concluded that there is no evidence for the compaction process in humans. Excessive proliferation of trabeculae was observed in mice after the introduction of a genetic variant of human LVNC resulting in the development of the LVNC phenotype [21,22]. This supports the idea that the LVNC phenotype and increased NC/C ratio are caused by the proliferation of trabeculae, but not as a result of compaction disturbances [6]. The hypertrabeculation was observed in humans as an adaptive phenomenon to increased cardiac output in athletes, pregnant women between the first and third trimester, and patients with hemoglobinopathy [6]. The physiological effect of hypertrabeculation remains controversial. The trabeculated ventricle can work more efficiently generating the same stroke volume at lower strain and wall stress. Moreover, hypertrabeculation can be reversible as seen in pregnant women, which supports its adaptive role. Hypertrabeculation observed in African athletes, however, together with a slight decrease in EF, was not a sign of pathological remodeling [12]. Kawel-Boehm et colleagues found decreased circumferential strain in patients with a higher degree of trabeculation [13]. The “theory of non-compaction”, however very attractive and convincing, is not confirmed in available publications. This does not necessarily mean that the theory is false. Perhaps we are dealing with a disease that has many etiologies, and the only common denominator is the excessive trabecular phenotype.

## 3. Clinical Presentation

The most common clinical presentations at the time of the first diagnosis are heart failure, thromboembolic events, and different rhythm disturbances. Among 241 patients from the German registry, 61% presented heart failure symptoms at the time of the first presentation. Thromboembolic events were present in 15% of patients [23]. 

Both atrial and ventricular arrhythmias are frequently observed. Incidence of atrial fibrillation varies from 15 to 23% and can cause systemic embolization. AF was more frequently observed in patients with reduced left ventricular systolic function [24]. Other common arrhythmias include supraventricular arrhythmias (4%), AV nodal reentry tachycardia (1%), and typical atrial flutter (1.5%). The WPW (Wolff–Parkinson–White) syndrome is more frequent in children (20%) than in adults, where it is present in about 1.5% [4]. 

Howard et al. found WPW in 11% among 348 children with LVNC [25]. The etiology of WPW syndrome among LVNC patients remains controversial. Ichida et al. suggest a disturbance in the development of fibrous annulus [26]; however, it was not confirmed by others [9]. 

Ventricular tachyarrhythmias are more common and occur in 47% of cases [27]. In the meta-analysis including 135 patients, ventricular tachycardia was found in 68% of them being polymorphic in 19%. The re-entry mechanism was identified in 83% of patients [28]. Ventricular fibrillation is a common cause of sudden cardiac arrest among patients with LVNC. Atrial fibrillation (AF), left bundle branch block, and impaired LV ejection fraction (LVEF < 35%) were at high risk for relevant clinical events. 

In patients with LVNC various, concomitant congenital heart diseases were reported, such as, for example, ventricular septal defect, atrial septal defect, patent ductus arteriosus Botalli, Ebstein’s anomaly, tetralogy of Fallot, pulmonary atresia, or bicuspid aortic valve. 

Many LVNC patients suffer from concomitant neuromuscular disorders or hereditary neuropathies such as Becker muscular dystrophy, Charcot–Marie–Tooth, Emery–Dreifuss muscular dystrophy, myotonic dystrophy, Leber’s hereditary optic neuropathy, or Barth syndrome [29]. Neurological examination performed in 220 patients with LVNC was normal only in 18% of them and the presence of neuromuscular disorders was a predictor of worse outcomes during follow-up [29].

Coronary artery disease is a rare finding in patients with LVNC. In addition to atherosclerotic etiology, a myocardial infarction caused by coronary artery embolization was reported [30]. Surgical treatment of coronary artery disease can be safely performed [31]. 

## 4. Diagnostic Methods

### 4.1. Echocardiography

Echocardiography is the most used diagnostic tool in patients with LVNC. The typical picture of isolated LVNC includes excessive trabeculation of the left ventricle, the best seen in the short axis or in the four-chamber view. The trabeculation often resembles a honeycomb, and when the space between trabeculae is narrow, it looks like a hypertrophied muscle. The use of color doppler or a contrast agent such as, for example, SonoVue, makes the recess between trabeculation visible and allows the diagnosis (Figure 1A–F). Among many proposed diagnostic criteria three seem to be mainly used (Table 2). The first was proposed by Chin et al. in 1990 after examining eight children [3]. They measured the distance from the deepest trabecular recess to the epicardial surface (labeled as X) and the distance from the tip of the trabeculation to the epicardial surface (labeled as Y). A ratio of X/Y ≤ 0.5 was considered pathologic suggesting the possibility of LVNC with a sensitivity of 79–100%, and specificity of 54–92%. The measurement should be performed in end-diastole in apical four-chamber view or parasternal long axis. Different diagnostic criteria were proposed by Stöllberger and colleagues [32]. The analysis was performed in end-diastole in an apical four-chamber view. The presence of at least three trabeculations protruding in the left ventricle apically from papillary muscle, having echogenicity such as myocardium, and synchronic movement during systole was considered pathologic. Moreover, there should be blood flow between trabeculations. Additionally, the compacted to non-compacted myocardium ratio should be >2 in end-diastole. Finally, Jenni et al. after analyzing 31 patients also proposed a ratio of NC/C myocardium > 2 but measured in a parasternal short axis end-systolic view. Additionally, three more criteria were necessary to establish the diagnosis: an absence of coexisting cardiac abnormalities and the presence of deep trabeculations that must be filled with blood [33]. The Jenni criteria are probably the most often used, but the comparison with two other methods showed the lowest reproducibility and diagnostic validity [34]. One must notice that the number of patients analyzed in all cited studies was extremely low. Some other echocardiographic criteria such as circumferential strain or left ventricular twist were proposed by other groups but the number of examined patients was also low [35,36]. Finally, the varying echogenicity and lack of a “golden standard” make the diagnosis very difficult. 

### 4.2. Magnet Resonance Imaging

Thanks to its ability to display anatomic details and functional information, cardiac magnetic resonance (CMR) has become the method of choice for diagnosing noncompaction cardiomyopathy [37]. Balanced steady-state free precession (SSFP) cine imaging and its high resolution allow very accurate assessment of the myocardium, in particular, differentiation of compact and non-compact myocardium (Figure 2 and Figure 3) [38]. Representation of the myocardium in different planes is the principal advantage of CMR. In addition, the possibility to visualize intracardiac thrombi is of clinical value (Figure 4A,B). The late gadolinium enhancement (LGE) helps rule out concomitant or underlying fibrosis (Figure 4C). The 3D approach allows taking the entire volume of the heart in each favorable view without the usual restriction of echocardiography [39]. Poor echocardiographic windows, difficulties with depicting the apex, and examiner dependency are known limitations of echocardiography and are not the case in CMR [39]. Therefore, the probability of missing segmental myocardial changes is significantly lower in CMR compared to echo. It should also be noted that CMR provides additional information about the mediastinum and thoracic structures [40]. In contrast, higher costs, relatively long acquisition times, and limited availability may make cardiac CMR a second-line imaging technique generally used after echocardiography. 

Typically, routing protocols in CMR include both cine SSFP images and black blood. It is important to compare both images simultaneously to avoid over or underestimation [41]. There is various literature in which criteria for LVNC have been described (Table 2).

The MRI diagnostic criteria proposed by Petersen et al. are generally the most accepted [42]. Petersen’s criteria were based on two approaches; first, on the long axis, and second, on the short axis. In both, segments 17 (AHA) and papillary muscle were excluded. An end-diastolic ratio between non-compacted and compacted layers greater than 2.3 is considered diagnostic of myocardial noncompaction with specificity (99%), sensitivity (86%), positive prediction (75%), and negative prediction (99%) [42]. Stacey et al. took a different strategy. They depicted LVNC only in the short axis with a diagnostic NC/C ratio of ≥ 2.0 in end-systole [43]. Jacquier et al. have used another method to measure end-diastolic global and trabeculated LV mass. They applied the following equation: (LV mass with trabeculations—LV mass without trabeculations)/(LV mass without trabeculations) × 100. Ruling in cut-off was >20% [44]. The same approach has been used by Choi et al. with a higher cut-off of >35% [45]. Grothoff et al. reported a trabeculated ventricular mass of more than 25% of the global LV mass and a non-compacted mass of more than 15 g/m^2^ as highly sensitive and specific for the diagnosis of LVNC [39]. The Grothoff criteria give also an opportunity to differentiate the LVNC from excessive trabeculation associated with thalassemia [49]. The noncompaction-to-compaction ratio > 2 measured in myocardial segments 4–6 suggests a high probability of LVNC. The maximal fractal dimension was used to diagnose LVNC by Captur et al. [46]. Global fractal dimension > 1.26 and apical fractal dimension > 1.3 were found in patients with LVNC. The measurements were performed in end-diastole. 

The ability of cardiac CMR to assess the right ventricle compared to echocardiography makes it a robust method for evaluating RV in suspicious NC cases. Unfortunately, there are fewer works available that deal with RV and NC. Because of the particular anatomy of the right ventricle, it is not easy to distinguish between pathologic and anatomically normal trabeculations. This right ventricle property leads to underestimating reported RV involvement in NC. [50] Otherwise, radiologic determination of a noncompaction-to-compaction ratio in the right ventricle is difficult (Figure 2E). Dilatation of the RV could be a supportive fact [51]. LGE has not yet been recognized as a final criterion for NC, but it provides dozens of crucial supporting information [52]. Because arrhythmia is the leading cause of death in LVNC patients, recognizing underlying fibrosis is very important [53]. A correlation between the amount of LGE and LVNC severity course was found. The patients with higher LGE were found to have more advanced diseases [54].

Our protocol for analysis of a patient with excessive trabeculation in CMR is as follows:

Retrospectively gated LV 2-, 3-, and 4-chamber long-axis (LAX) single-slice and short-axis oblique (SAO) base-to-apex stack cine bSSFP sequences with 6–8 mm slice thickness, 2 mm slice gap, pixel size ≤ 1.8 × 1.8 mm^2^, and a temporal resolution < 50 ms.

A typical examination procedure for non-compaction patients includes 2-chamber (1 slice), 3-chamber (3 slices), and 4-chamber (3 slices) long-axis views as well as a short-axis stack covering the entire left/right ventricle (12–16 slices, no gap) using cine steady-state free precession acquisitions (TR/TE/flip angle = 2.7 ms/1.35 ms/42°) to assess cardiac function and morphology. Spatial resolution should be approximately 1.5 × 1.5 × 8 mm³ and at least 30 cardiac phases per cardiac cycle to achieve an acquisition time of 30–35 ms per cardiac phase (assuming a heart rate of approximately 60 beats per minute). Parallel imaging techniques should restrict breath-hold periods to less than 12 s (Figure 2 and Figure 3).

How do we measure?

1. We find the maximum manifestation of trabeculae in the short and long axes in end-diastole. If the non-compaction to compaction ratio > 2.3, consider the diagnosis of LVNC. 

2. We calculate trabeculated mass including the blood pool at end-diastolic. A percentage of >20% of trabeculated mass confirms the diagnosis (Figure 5). 

Regardless of the decision for which strategy, whether to measure at end-diastole or end-systole remains controversial. 

### 4.3. ECG

Despite complex alterations in the structure of heart tissue, there are no specific changes in 12-lead ECG examinations. The most frequent findings are left bundle branch block (LBBB), ST changes including T wave inversion, pathologic Q waves, ventricular ectopies, and AV blocks. According to Mavrogeni et al. ECG changes are present in up to 90% of cases [55]. Fragmentation of QRS, especially R wave was present in 47% of patients (Figure 6) and correlated with myocardial fibrosis and increased risk of sudden cardiac death [56,57]. According to Steffel et al. who analyzed 78 patients with a diagnosis of isolated left ventricular noncompaction, the most frequent ECG changes were repolarization abnormalities (72%), including prolongation of QTc in 52% of patients. The signs of left ventricular hypertrophy were present in 38% and LBBB in 19% of patients. Only 13% of patients had normal ECGs [58]. 

### 4.4. Computer Tomography

The multidetector computed tomography (MDCT) can detect non-compacted myocardium and at the same time offers an outstanding representation of the anatomy of the heart and coronary arteries. However, as compared to CMR, it is not able to characterize myocardial fibrosis, which is an important prognostic factor. The only criteria to diagnose LVNC with MDCT were proposed by Melendez-Ramirez et al. Finding an NC/C ratio > 2.2 in at least two or more segments, provided a correct diagnosis with 100% sensitivity and 95% specificity (Table 2) [48]. In addition, is not to forget, that the generation of cine images in a full-cycle CT examination is connected with radiation exposure. That is, why a CT examination to rule out LVNC, is only an option if there are contraindications for CMR [59].

### 4.5. Endomyocardial Biopsy

There are very limited data available concerning the endomyocardial biopsy’s ability to diagnose VNC. The available data showed different pathologies; however, it is not possible to differentiate the isolated from the non-isolated form of LVNC [60]. The endomyocardial biopsy can be useful in detecting the secondary causes of LVNC, especially in patients with concomitant neuromuscular diseases. The indications to perform the endomyocardial biopsy should be determined carefully on an individual basis. 

## 5. Genetics

LVNC is a rare genetic form of cardiomyopathy. In the German TORCH registry [61] and the European EUROP cardiomyopathy registry, 5% or 4.1% of cases were diagnosed as LVNC patients, respectively. Recently, in a large retrospective study 327 unrelated LVNC were tested for disease-related DNA variants. About 66 genes were considered to be associated with LVNC, 82% of those are coding for sarcomeric proteins [62]. Of note, in pediatric LVNC-patients, gene variants classified (probably) pathogenic (ACMG 4 or 5) were more frequently found than in adult LVNC. The genes associated with LVNC show considerable overlap with other forms of cardiomyopathies and especially with DCM and HCM. In a systematic meta-analysis, van Waning et al. evaluated 66 genes in 561 patients from 172 studies [11]. They found that 52% of affected genes are coding for sarcomeric proteins, and 9% for arrhythmia-associated genes. Non-sarcomeric, mitochondrial, and complex genotypes were found in 6–8% of cases, respectively. The most frequently affected genes were *MYH7*, *MYBPC3,* and *TTN*. In pediatric cases of LVNC, the genetic background of the disease is frequently associated with complex congenital syndromes, mitochondrial diseases, chromosomal defects, or X-linked. Disease-related variants in the genes *MYBPC3*, *TTN*, arrhythmia genes, or X-linked genotypes are associated with a high risk for major adverse cardiac events [11]. Although the correlation of genetic variables with the clinical outcome is impressive, little is known about the pathomechanisms of the variants. In addition, the penetrance of variants in cardiomyopathies is frequently incomplete and dependent on the noncompaction phenotype. Complex genotypes are rare. The highest degree of co-segregation was found in isolated LVNC and cases with the combined phenotype of DCM and LVNC [63]. This underscores that genetic testing in LVNC is of clinical relevance and should be considered.

## 6. Differential Diagnosis

Improvement and accessibility of imaging techniques increased the number of detected patients with prominent trabeculation, which does not necessarily mean pathology. The most difficult problem in the differential diagnosis of LVNC is distinguishing pathological hypertrabeculation from physiological, resulting, for example, from individual or ethnic differences. A misdiagnose can be connected with the danger of stigmatizing healthy people. The diagnostic criteria are ambiguous and depend greatly on the method and criterion used. Alone, the most used NC/C ratio can be easily affected by hypertrophy of the compacted layer. 

First of all, the proper identification of left ventricular structures such as LV thrombus, false tendons, aberrant chords, cardiac fibromas, eosinophilic heart disease, endomyocardial fibrosis, and cardiac metastasis, which can imitate LVNC, should be performed. 

A multidisciplinary approach including a cardiologist, imaging specialist, geneticist, or rhythmologist is necessary to properly select the affected patients. Different diagnostic strategies have been proposed; however, none are generally accepted. Adabifirouzjaei and colleagues proposed the valuation of the left ventricular function in patients in whom the hypertrabeculation was found on echocardiography or CMR [64]. In patients with normal EF and no associated conditions, a family screening of first-degree relatives should be performed. No evidence of familial involvement makes diagnosing a normal variant of LVNC likely. In the case of familial involvement, the genetic origin of LVNC should be taken into consideration. In the case of impaired EF, the authors propose two possibilities: LVNC is caused by a genetic disorder and the LV dysfunction is secondary to LVNC. The second is when LVNC is caused by LV remodeling, for example, in DCM or HCM. Unfortunately, this theoretical approach is not quite sufficient to guide the diagnostic procedure in a real clinical situation. A much more practical approach was proposed by D’Silva [6]. In patients in whom hypertrabeculation was found, a clinical examination, ECG, family history, and CMR or echocardiography should be performed. In the case of isolated LVNC without any other pathologies and negative family history, physiological hypertrabeculation would be considered. The probability of a disease is low. In the opposite case, family screening and testing including genetic testing, ambulatory ECG monitoring, and exercise testing should be performed. If the abnormalities consistent with cardiomyopathy were found without evidence of other cardiac diseases, LVNC cardiomyopathy should be diagnosed. The patients without clear signs of cardiomyopathy should be followed up as the exclusion of the disease is not possible. 

Our policy (Figure 7) depends on whether the LVNC was found accidentally or the patient presented with symptoms. In asymptomatic patients, family history is analyzed, a 12-lead ECG is performed and a supplementary echo or CMR is made, depending on which test was performed at the beginning. The value of CMR is not only to confirm hypertrabeculation, but also to confirm or exclude the presence of late gadolinium enhancement (LGE) and thrombi, and to assess the function of the left ventricle. Myocardial fibrosis and reduced EF were both found to be related to a worse prognosis. Patients with no confirmed pathology have a shallow risk of cardiac events and genetic testing and close follow-up are not necessary. It is essential to look for cases with adaptive or transient hypertrabeculation. 

In symptomatic patients, a complete cardiological examination should be performed including 12-lead ECG, 24 h ECG monitoring, echocardiography, and CMR with special attention paid to LGE and thrombi. The family history of three generations should be carefully analyzed. All first-degree family members should undergo a complete cardiological examination including echocardiography and CMR. Particular attention should be paid to coexisting congenital anomalies such as neuromuscular disorders and congenital heart diseases. Genetic testing should be performed in every case using gene panels looking for cardiomyopathies identified to be associated with LVNC. The index patient of the family should be tested at the beginning, and in the case of positive results, the family should be offered genetic counseling and testing [65]. 

## 7. Treatment

LVNC has no specific therapy as of today. In the actual ESC guidelines for the treatment of heart failure, no special recommendations for the management of LVNC are given. Treatment should be applied depending on the clinical situation and specific indications. In patients with decreased LV EF, heart failure pharmacotherapy should be applied according to the guidelines [66]. In patients with end-stage heart failure refractory to medical therapy, heart transplantation should be considered. In patients with contraindication to heart transplantation, implantation of a left ventricular assist device can be an effective therapy option. Recently, Takamatsu and colleagues reported a successful surgical resection of non-compacted myocardium in a 65-year-old man with the improvement of systolic function of the left ventricle after a one-year follow-up [67]. 

By the current guidelines of ESC, there is no special indication for ICD implantation for patients with LVNC. Implantation of ICD should be considered in all patients with LV EF lower or equal to 35%, as in patients with dilatative cardiomyopathy. Implantation of an ICD results in cardiac arrest prevention and improved prognosis. Appropriate ICD therapy was reported in 67% of patients [66]. 

In patients with LVNC and advanced heart failure who fulfill general criteria for resynchronization therapy (CRT), implantation of CRT should be considered. In responders, implantation of CRT in overall improvement of the ejection fraction and reverse remodeling of heart muscle tissue was reported [68,69]. Recurrent arrhythmias may be treated with transcatheter ablation. Muser et al. and Sohns et al. reported good results in the ablation of VA substrates in patients with LVNC and VT with good VA control in long-term follow-up [70,71].

Ablation may also be considered in the case of atrial fibrillation; however, no long-term follow-up data are available. Ablation may be considered as well in other co-existing with LVNC abnormalities, such as WPW. 

According to the study by Chimenti and colleagues, thromboembolic complications in patients with LVNC can reach 38% [72]. In the retrospective analysis of 169 patients, thromboembolic complications were noted in 15% [23]. The cardiac source of embolism was found in 69% of patients; however, atrial fibrillation was found in only 39% of them. Excessive trabeculation of the left ventricle was found to be a risk factor for thromboembolic risk. A 5% increase in the volume of non-compacted myocardium was associated with a ninefold risk of thromboembolic events [73]. 

In the prevention of embolic events, both VKA (vitamin K antagonists) and NOAC (non-vitamin K antagonist oral anticoagulant) drugs may be considered. In the group of patients with LVNC and coexisting atrial fibrillation or a history of embolic events, NOAC is the first choice. In patients with left ventricular thrombus, VKA are the drugs of choice. There is no agreement as to whether patients with left ventricular dysfunction without previous history of atrial fibrillation should be treated with anticoagulants. In the randomized WARCEF study 2305 patients with EF < 35% and sinus rhythm were randomized to Warfarin (INR 2.0–3.0) and Aspirin 325 mg and followed up for 6 years. The primary outcome was the time to the first event in a composite endpoint of ischemic stroke, intracerebral hemorrhage, or death from any cause. No significant overall difference between the two treatments could be found. Warfarin was statistically more effective in the reduction in ischemic stroke (0.72 events per 100 patient-years vs. 1.36 per 100 patient-years; hazard ratio, 0.52; 95% CI, 0.33 to 0.82; *p* = 0.005); however, the incidence of major hemorrhage was significantly higher in the Warfarin group what offset its beneficial effect [74]. A negative correlation of risk of embolic events with EF was found [75]. The decision should be taken individually considering individual thrombosis and hemorrhagic risk. 

In patients with normal LVEF, the decision can be made by taking into consideration the CHADS_2_ score. Chimenti et al. proposed starting treatment with NOACs in patients with a CHADS_2_ score > 2 [72]. 

Because many cases of LVNC occur in a familial manner, the screening of patient relatives should be performed. Such screening should include genetic tests to confirm the disease and future risk prediction. Due to the overall lack of specific traits of LVNC and specific tests allowing confirmation of diagnosis, a follow-up seems to be the best strategy in therapy. The regular patient examination allows for proper treatment and new data collection on this complex disorder.

According to ESC recommendations for participation in competitive sports [76], athletes with LVNC, with excessive trabeculations revealed in any imaging method, without impaired LV systolic function, ventricular arrhythmias, or thromboembolic events, have no restrictions in competitive sports (Class IIa/Level B) with the exception of sports where possible syncope may cause harm or death (Class IIb/Level C). However, in patients with impaired LV systolic function, ns-VT is recommended to abstain from sports on a competitive level. Limiting exercise levels to leisure-time activities with systematic clinical observation is advised (Class III/level C).

## 8. Prognosis

Given the fact that there is no uniform definition of LVNC and that there are different criteria for diagnosis, the prognosis of the course of the disease is difficult, if not impossible. The situation is additionally complicated by the fact that the studies available in the literature concern various cohorts of patients diagnosed with the use of various criteria. 

For the reasons mentioned above, the mortality of patients diagnosed with LVNC in follow-up could not be unequivocally assessed. In a meta-analysis of 1822 patients performed by Aung, the event rate for cardiovascular mortality was 1.92 (95% CI, 1.54–2.30) per 100 person-years during 2.9 years of follow-up [77]. In comparison, the event rate in a general population was 0.08 and 0.41 per 100 person-years for cardiovascular and all-cause mortality. The risk of cardiovascular mortality was similar to patients with dilatative cardiomyopathy. The extent of trabeculation had no significant impact on mortality, but the worse survival was noted in patients with impaired left ventricular function. 

In the MESA (Multi-Ethnic Study of Atherosclerosis) 2742 patients with LVNC were followed for 9.5 years [78]. The diagnosis was made with CMR using Petersen criteria. The authors analyzed the end-systolic volume index (ESVi), end-diastolic volume index (EDVi), and EF during follow-up depending on the extent of trabeculation. No clinically relevant changes in EDVi, ESVi, or EF were found irrespectively from the extent of trabeculation. 

A benign course of the disease also in symptomatic patients was found by Murphy and colleagues who followed 45 patients with LVNC for more than 10 years. A large number of patients were symptomatic at presentation: 62% reported dyspnea, 91% had abnormal ECG, and 66% impaired left ventricular function [79]. Freedom from death or transplantation was 97% after 46 months despite a highly symptomatic cohort.

The retrospective analysis of 339 patients with LVNC performed by the Mayo group showed reduced survival of patients with LVNC compared to that expected in the US population over 6.3 years [80]. LVNC was defined using Jenni, Chin (echocardiography), and Petersen (CMR) criteria. The authors found higher mortality in patients with LV EF < 50% and the noncompaction zone extending from the apex to mid or basal segments of the left ventricle as compared to the general US population (*p* < 0.001). The differences were not found in patients with localized apical trabeculation and with preserved LV EF. The survival in this group of patients was similar to the general US population. 

Many other parameters were found to correlate negatively with the prognosis as left atrial size, sustained ventricular arrhythmias, atrial fibrillation, or left bundle branch block. There are no generally accepted parameters allowing for long-term prognosis in patients with LVNC.

## 9. Conclusions

We are dealing with a disease that has two different definitions. Pathogenesis is unclear, and incidence remains difficult to define because of inconsistent diagnostic criteria and the lack of a “golden standard”. Available observational studies deal with different patient cohorts diagnosed with different modalities, with the use of different criteria. Many different terms are used to describe a “noncompaction” phenotype, which makes a comparison of studies more difficult or simply impossible. Estimating the prognosis under these conditions is also uncertain. The biggest problem we face in daily practice is identifying the patients with isolated LVNC and patients with potentially reversible hypertrabeculation. A mistake leads to unnecessary examinations, costs, and most importantly, sentencing healthy people to live with the diagnosis of the potentially deadly disease. On the other hand, delay in treatment can have serious consequences. Unfortunately, at present, we do not have examination techniques that would allow us to distinguish these groups of patients with sufficient accuracy. That is why diagnosing LVNC using a single modality only should be avoided. A multidisciplinary approach including family history and genetic testing is essential to identify a true disease and should be performed in every case. An individual approach is necessary for the treatment of symptomatic patients.

## Figures and Tables

**Figure 1 jcm-11-04135-f001:**
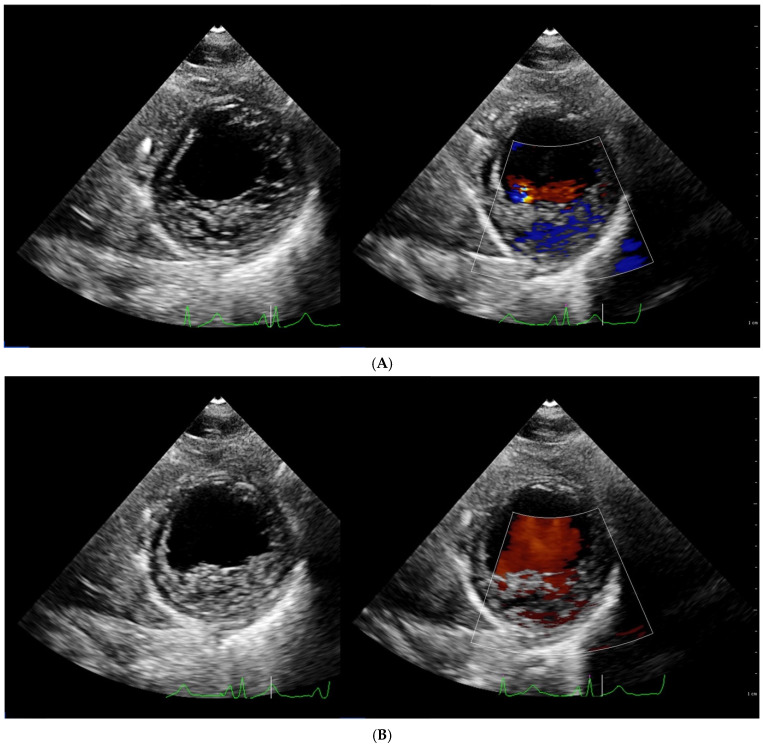
Transthoracic echocardiographic examination, patient with LVNC. (**A**) Parasternal short axis, 2D, diastole, right with color doppler. (**B**) Parasternal short axis, 2D, systole, right with color doppler. (**C**) Apical 4-chamber view, 2D, diastole. Left—note excessive trabeculation of the left ventricle. The white arrow shows the deep trabecular recess between the trabeculation. Right—healthy patient with a normal left ventricle. (**D**) Zoomed view of the same patient, apex, and apical segments of the lateral wall. Note excessive trabeculation of the left ventricle. The white arrow shows the deep trabecular recess between the trabeculation. (**E**) Zoomed view with color doppler of the same patient, apex, and apical segments of the lateral wall. Note excessive trabeculation of the left ventricle filled with blood as seen with color doppler (arrows). (**F**) Contrast echocardiography (SonoVue), modified apical 4 chamber view, end-diastole, patient with LVNC. Note the deep trabecular recess between trabeculation of the left ventricle filled with contrasted blood.

**Figure 2 jcm-11-04135-f002:**
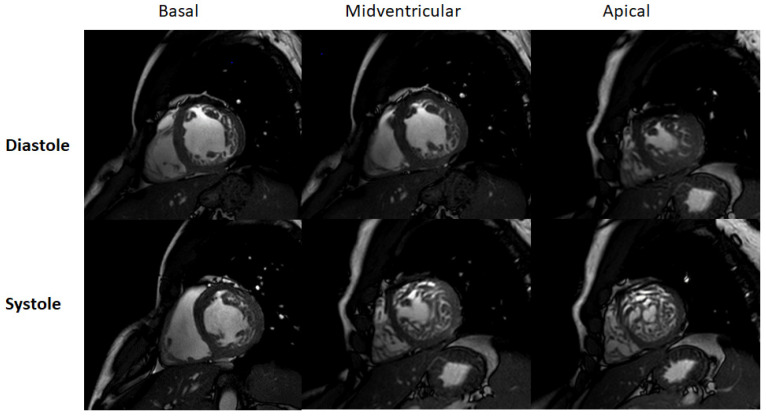
CMR examination. 25 years old male. Basal, midventricular, and apical cine steady-state free precession images in short-axis orientation during end-diastole and end-systole. Note differentiation between non-compacted from compacted myocardium.

**Figure 3 jcm-11-04135-f003:**
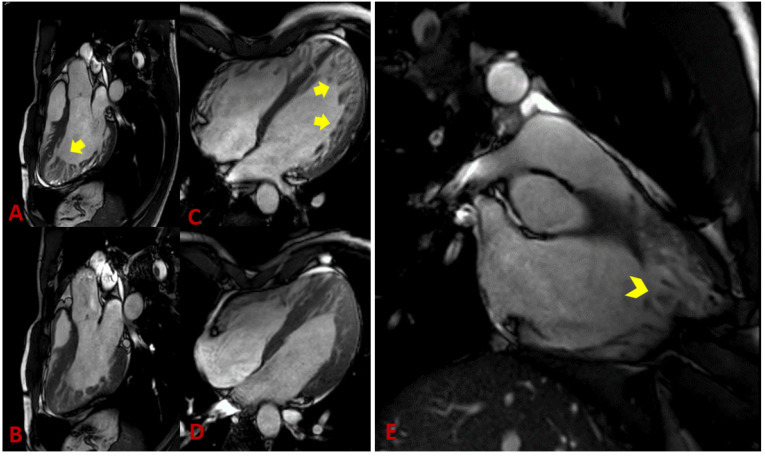
CMR examination: (**A**) Three-chamber view, diastole; (**B**) three-chamber view, systole; (**C**) four-chamber view, diastole; (**D**) four-chamber view, systole. Cine steady-state free precession images at end-diastole and end-systole. Note an increased number of trabeculations along the LV lateral wall and LV apex (arrows). (**E**) Long-axis right ventricular view, cine steady-state free precession image of the right ventricle showing prominent hypertrabeculation (arrowhead).

**Figure 4 jcm-11-04135-f004:**
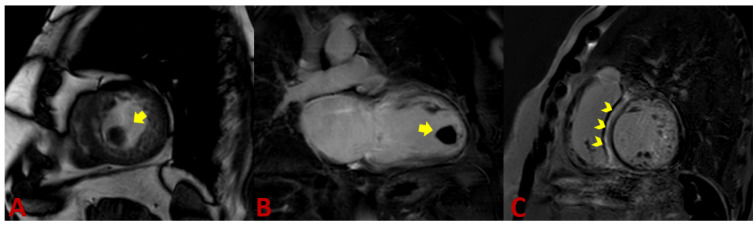
CMR examination: Thrombus in the apex of the left ventricle in a patient with LVNC (**A**,**B**) (arrow). (**C**) Short-axis view, late-enhanced sequence (PSIR). Extensive intramyocardial LGE in the septum (arrowheads).

**Figure 5 jcm-11-04135-f005:**
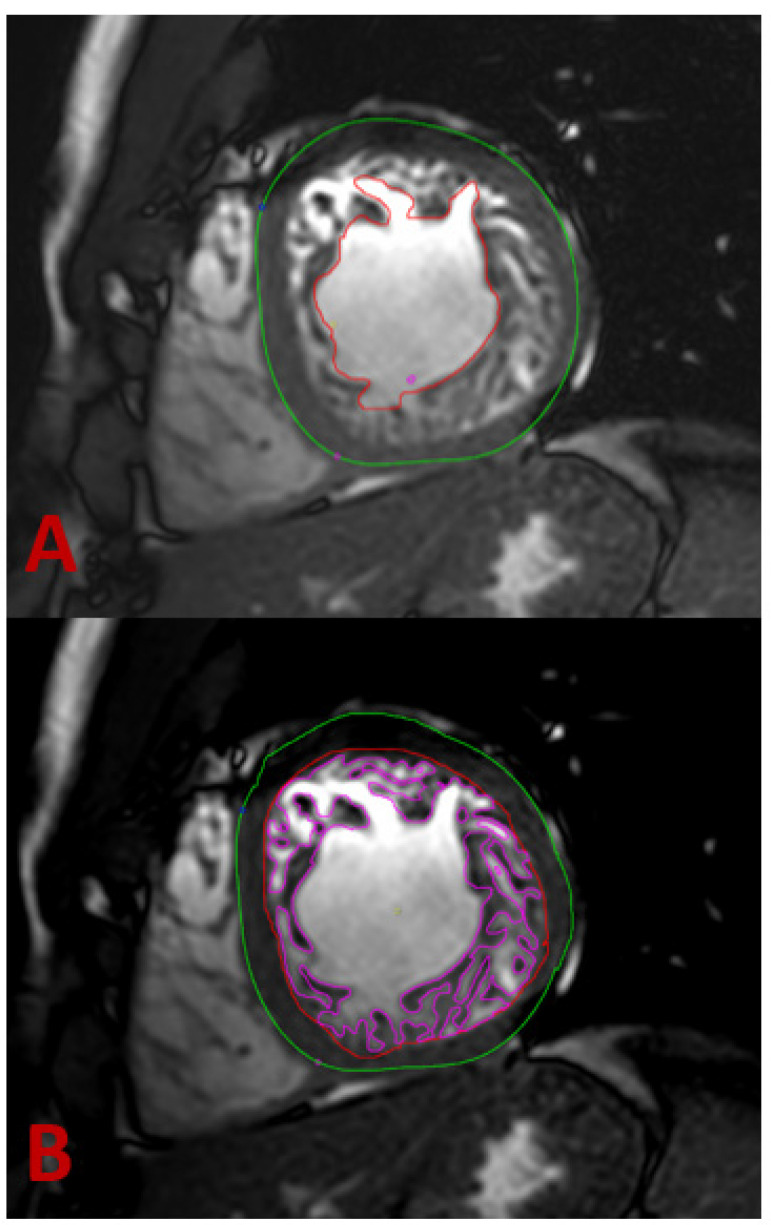
CMR examination, short-axis cine bSSFP image of the mid-cavitary LV. Prominent hypertrabeculation in a 20-year-old male with a history of ADPKD and LVNC phenotype. Measurement of the whole myocardial mass: (**A**) Determination of epicardial contours and endocardial contours at the NC myocardium border for measurement of whole myocardial mass. (**B**) Determination of endocardial contours at the border of the NC myocardium and non-trabeculated cavity for measurement of non-compacted myocardium mass. bSSFP—balanced steady-state free precession; LV— left ventricle. ADPKD—autosomal dominant polycystic kidney disease.

**Figure 6 jcm-11-04135-f006:**
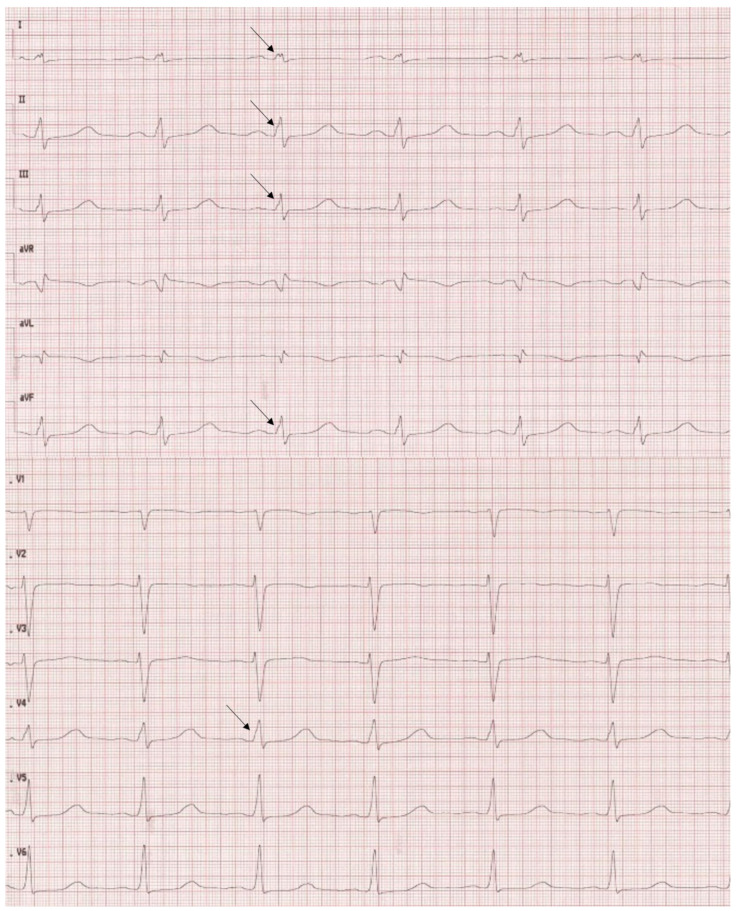
An Electrocardiogram of the patient from Figure 1C–E shows regular sinus rhythm 75/min. PQ 190 ms, QRS 100 ms, QT 460 ms, QTc 514 ms. Note notching of the R-wave of the QRS in lead I, II, III, aVF, and V4 (arrow) as well as prolongation of QTc.

**Figure 7 jcm-11-04135-f007:**
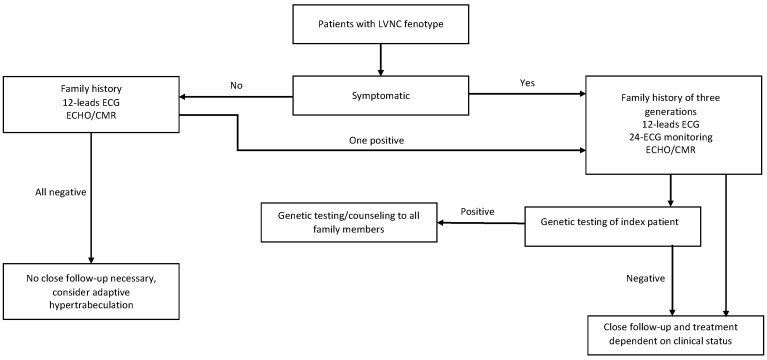
A pathway for risk stratification and follow-up of patients with hypertrabeculation phenotype.

**Table 1 jcm-11-04135-t001:** Subtypes of NCCM, modified after Arbustini et al. [10].

1. iLVNC. NC morphology in left ventricles with normal systolic and diastolic function, size, and wall thickness;
2. LVNC with LV dilation and dysfunction at onset, such as in the paradigmatic infantile CMP of Barth syndrome
3. LVNC in hearts fulfilling the diagnostic criteria for DCM, HCM, RCM, or ARVC;
4. LVNC associated with congenital heart disease;
5. Syndromes with LVNC, either sporadic or familial, in which the noncompaction morphology is one of the cardiac traits associated with both monogenic defects and chromosomal anomalies, i.e., complex syndromes with several multiorgan defects;
6. Acquired and potentially reversible LVNC, which has been reported in athletes; it has also been reported in sickle cell anemia, pregnancy, myopathies, and chronic renal failure;
7. Right ventricular noncompaction, concomitant with that of the left ventricle, or present as a unique anatomic area of NC.

**Table 2 jcm-11-04135-t002:** LVNC diagnostic criteria.

Author	Method	Diagnostic Criteria	Cardiac Phase	Cut-Off
**Petersen [42]**	CMR	Ratio of compacted epicardium and non-compacted endocardium	End diastole	NC/C ≥ 2.3
**Stacey [43]**	CMR	Ratio of compacted epicardium and non-compacted endocardium	Short axis, end-systole	NC/C ratio of ≥ 2.0
**Jacquier [44]**	CMR	A value of trabeculated LV mass above 20% of the global mass of the LV	End diastole	LV trabeculated mass > 20%
**Grothoff [39]**	CMR	Ratio of total LV trabeculated mass to global myocardial mass	End systole	Trabeculated ventricular mass greater than 25% of the global left ventricular mass; noncompacted mass greater than 15 g/m^2^
**Choi [45]**	CMR	A percentage of trabeculated myocardial volume of the total myocardial volume of the LV	End-diastole, long-axis	LV trabeculated volume > 35%
**Captur [46]**	CMR	Maximal fractal dimension	End diastole	Global fractal dimension > 1.26; apical fractal dimension > 1.3
**Chin [3]**	ECHO	The ratio of the distance from the deepest trabecular recess to the epicardial surface (X) and the distance from the tip of the trabeculation to the epicardial surface (Y)	Long axis, end diastole	X/Y ≤ 0.5
**Stöllberger [47]**	ECHO	Ratio of compacted and non-compacted endocardium. Presence of at least 3 trabeculations protruding in the left ventricle apically from papillary muscle, presence of the blood flow between trabeculations.	Four chamber, end diastole	NC/C > 2
**Jenni [33]**	ECHO	Ratio of compacted and non-compacted endocardium. Absence of coexisting cardiac abnormalities, the presence of deep trabeculations, which are filled with blood	Short axis, end systole	NC/C ≥ 2
**Melendez-Ramirez [48]**	MDCT	Ratio of compacted and non-compacted endocardium in at least 2 or more segments	All 17 segments, end diastole	NC/C ratio > 2.2

## Data Availability

Not applicable.

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
