# Peer review of "Left Ventricular Non-Compaction Cardiomyopathy-Still More Questions than Answers"

_jcm, 2022, doi:10.3390/jcm11144135_

Round 1
Reviewer 1 Report
The manuscript is well written. The authors analyze several criteria used in the diagnosis of Left ventricular non-compaction cardiomyopathy. Authors should correct the cardiac phase in table 2 regarding the Petersen criterion (no end-systole but end-diastole).
In addition, the authors could add more information about the importance of Grothoff's criterion in differentiating Left ventricular non-compaction cardiomyopathy from other forms of hypertrabeculation such as those present in thalassemia. In this sense the authors could add and cite the works of Macaione F and Pepe A on this topic.
Author Response
Dear Reviewer 1
Thank you for your review and for your valuable comments.
We corrected the cardiac phase in table 2 regarding the Petersen criterion.
Following your suggestion, we add some more information about Grothoff's criterion.
Yours faithfully,
J. Paluszkiewicz
Reviewer 2 Report
Paluszkiewic et al. have written a detailed account of the left ventricular non-compaction. They summarize the current state of the art in the pathogenesis, clinical manifestation, diagnosis, treatment, and prognosis of the LVNC. The authors particularly emphasized the difficulties arising from the non-uniform classification and diagnostic criteria. Overall, this is a didactic and useful manuscript with many references. However, attention should be paid to the following points.
1. To be complete the authors should add a scheme proving the graphical information about the normal and LVNC heart morphology.
2. It would also be helpful to have a brief definition of the LVCN just at the beginning of the article.
3. Data on differences in ethnic prevalence should be included.
4. The authors mentioned two accepted hypotheses about LVNC. The authors could consider adding more evidence for each of them in the manuscript.
5. Since the part of the coronary vessels as well as Purkinje fibers as the terminal part of the cardiac conduction system have the origine in the trabeculae is it known if these structures are functionally affected in the LVNC?
5. The summarization of diagnostic criteria, as well as provided images, are useful. However, it is not clear what „labeled as X“ means. The image showing this parameter should be helpful. Also, both echocardiographic and MRI images would benefit from a more detailed description depicting the hypertrabeculation.
6. On page 4, line 140. „The hypertrabeculation was observed in humans as an adaptive phenomenon to increased cardiac output in athletes, pregnant women between the first and third. The trabeculated ventricle can work more efficiently generating the same stroke volume at lower strain and wall stress.“ On the other hand, is generally accepted that hypertrabeculation has a deleterious effect on cardiac pumping efficiency. Do the authors elaborate on this discrepancy?
7. On page 5, the authors provide the association of LVNC with different rhythm disturbances. Could authors provide commentary about the connection between the atrial and supraventricular types of arrhythmias and LVNC? Does it mean that hypertrabeculation affects the atrial tissue as well?
8. On page 5, the abbreviation for WPW syndrome is not explained. Does it mean that the hypertrabeculation is connected with the persistence of the accessory pathway?
9. Table 2 should include the direct reference.
10. On page 17, „repolarization abnormalities (72%), including prolongation of QTc in 52% of patients.“ Are they known mutations of specific ion channels related to the LVNC which provide substrate for long QT?
11. On page 18, there is no figure legend to the ecg image.
12. The diagnostic methods are well described, the review will further benefit from including the differential diagnostic scheme.
13. The abbreviations VKA and NOAH are not explained.
Author Response
Dear reviewer 2.
Thank you very much for your comments. We modified our manuscript according to your comments.
- We modified Fig. 1C adding a picture of a normal left ventricle to make the differences more readable.
- We added a brief definition of the LVCN at the beginning of the article.
- We complete the data on differences in ethnic prevalence.
- Following your suggestion we added additional citations.
- The frequent occurrence of both atrioventricular and intraventricular conduction disturbances in patients with LVNC suggests that there is an association between LVNC and conduction disturbances. Morphogenesis disorders of the conduction system is integrally related with ventricular maturation potentially explaining conduction defects associated with congenital malformations or inherited cardiomyopathies.
5.1. We modified the description in the table 2, also concerning Chin criteria. We hope this will improve the readability of the table. We consciously gave up the graphic representation of individual criteria (we included 10 in the table) because it would mean introducing many additional figures.
- Following your proposal we tried to explain that actually the physiological effect of hypertrabeculation remains controversial and no clear answer can be given.
- The cause of the frequent occurrence of supraventricular (and also ventricular) arrhythmias is not fully established. The authors do not know a case of the atria being occupied by the non-compaction process. In the the development process of the embryonic heart the ventricles and atria develop from different segments of the primary linear heart tube. The cardiac arrhythmias can be caused by many different secondary factors like for example cardiac chamber enlargement, pressure or volume overload.
- We explained the abbreviation for WPW. The possible connection of WPW and LVNC has not be proofed yet, and existing data are controversial. We modified the manuscript following your suggestion.
- We added the references to table 2 and additionally modified the descriptions hoping to make it more readable.
- Among patients with the left ventricular noncompaction 66 different genes were identified, among them also genes responsible for long QT syndrom like : KCNE1, KCNQ1, KCNH2, SCN5A and ANK2.
- We added the figure legend.
- We add the diagnostic scheme according to your suggestion.
- We explained the abbreviations of VKA and NOAC.
Yours faithfully,
J. Paluszkiewicz